# CRIF1-CDK2 Interface Inhibitors Enhance Taxol Inhibition of the Lethal Triple-Negative Breast Cancer

**DOI:** 10.3390/cancers14040989

**Published:** 2022-02-16

**Authors:** Xiaoye Sang, Nassira Belmessabih, Ruixuan Wang, Preyesh Stephen, Sheng-Xiang Lin

**Affiliations:** Axe Molecular Endocrinology and Nephrology, CHU de Quebec-Research Center (CHUL), Laval University, 2705 Boulevard Laurier, Quebec City, QC G1V 4G2, Canada; xiaoye.sang.1@ulaval.ca (X.S.); nassira.belmessabih.1@ulaval.ca (N.B.); ruixuan.wang.1@ulaval.ca (R.W.); stpriyesh@gmail.com (P.S.)

**Keywords:** triple-negative breast cancer, chemotherapy resistance, CR6-interacting factor 1, cyclin-dependent kinases 2, interface inhibitors

## Abstract

**Simple Summary:**

This study reported our most recent results for targeting triple-negative breast cancer (TNBC) with a low survival rate, using CR6-interacting factor 1–cyclin-dependent kinase 2 (CRIF1–CDK2) interface inhibitors, by inhibiting the resistance to taxol treatment. Presently, over 50% of TNBC patients become resistant to chemotherapy and, to date, no solution is available. The combined treatment, using CRIF1–CDK2 interface inhibitors with chemotherapy, provides an unprecedented strategy against the deadly TNBC.

**Abstract:**

Paclitaxel (taxol), a chemotherapeutic agent, remains the standard of care for the lethal triple-negative breast cancer (TNBC). However, over 50% of TNBC patients become resistant to chemotherapy and, to date, no solution is available. CR6-interacting factor 1 (CRIF1) is reported to act as a negative regulator of the cell cycle by interacting with cyclin-dependent kinase 2 (CDK2). In our study, two selective CRIF1–CDK2 interface inhibitors were used to investigate whether they could exert anti-proliferative activity on the TNBC cell lines. When combined with taxol treatment, these two inhibitors can advance the cells from G0/G1 to S and G2/M phases, producing irreparable damage to the cells, which then undergo apoptosis. Moreover, they enhanced the reduction in cell proliferation induced by taxol in TNBC cells, thereby improving sensitivity to taxol in these cell lines. Importantly, the inhibitors did not regulate the cell cycle in normal cells, indicating their high selectivity towards TNBC cells. Overall, the resistance to the anti-proliferative effects induced by taxol can be significantly reduced by the combined treatment with selective CRIF1–CDK2 interface inhibitors, making a conceptual advance in the CDK-related cancer treatment.

## 1. Introduction

Approximately 15–20% of breast cancers (BCs) can be classified as triple-negative breast cancer (TNBC) [1]. The absence of human epidermal growth factor receptor 2 (Her2) and both the estrogen and progesterone receptors are characteristic of this disease. This presents a major therapeutic challenge, due to the absence of a suitable endocrine therapy or effective target therapy [2,3]. Therefore, surgery and chemotherapy, either individually or in combination, remain the only available treatment interventions [4]. Paclitaxel (taxol), belonging to the class of Taxanes, remains the standard treatment for TNBC [5]. Taxol leads to mitotic catastrophe by stabilizing microtubules (MTs) and inhibiting their disassembly during metaphase, thus resulting in mitotic arrest and cell death [6,7]. Although the benefits of chemotherapy are clear, over 50% of TNBC patients become resistant to chemotherapy, typically after 6–10 months of treatment [8].

Particular serine/threonine kinases, known as cyclin-dependent kinases (CDKs), perform key functions by modulating transcription, in response to a number of extra- and intracellular cues, and by regulating cell division [8]. Abnormally activated CDKs result in increased cancer cell proliferation and genomic instability [9]. Thus, it has been suggested that inhibiting CDK activity may be an approach to treat various cancer forms, such as using WEE1. In breast cancer, the most extensively studied kinases are CDK4 and CDK6, their interaction with cyclin D can result in cell cycle transition from the G0 to G1 phase [10]. When used in combination with endocrine therapy, palbociclib, a CDK4 and CDK6 inhibitor, resulted in improved progression-free survival in ERα+/Her2-negative advanced breast cancer [11,12,13]. However, palbociclib showed no reduction in cell proliferation in the in vitro studies with TNBC [11,14].

CR6-interacting factor 1 (CRIF1) is a component of the mitochondrial ribosome large subunit. It is involved in a number of physiologic processes in humans and was first identified as a negative regulator of the cell cycle. Previous studies indicated that CRIF1 may have a regulatory involvement in microenvironment-induced leukemia cell cycle arrest by acting as a CDK2 inhibitor [15]. Protein–protein docking studies showed that two potential binding sites on CRIF1 are involved in the accommodation of CDK2: a small and long α helix [16]. It is suggested that the CRIF1–CDK2 interface inhibitor could improve cellular radio sensitivity in the osteosarcoma (OS) cell lines by selectively enhancing arrest in the G2/M phase and apoptosis associated with CDK2 overactivation in these cells [16]. On the other hand, although recent evidence points towards key negative regulators of CDKs as therapeutic targets in TNBC, a number of CDK inhibitors have undergone evaluation in clinical trials, but yielded disappointing results because of their low specificity to target specific CDK enzymes [3,17]. Therefore, more specific, and potent inhibitors, targeting CDK enzymes, are urgently needed for the treatment of TNBC.

The principal aim of this study was to investigate the effects of two selective CRIF1–CDK2 interface inhibitors, F1142-3225 and F0922-0913, to decrease the resistance to anti-proliferative effects induced by taxol in the TNBC cell lines MDA-MB-231 and MDA-MB468. Our study indicated that these two inhibitors promoted the transition of cells from G0/G1 to S phase and G2/M, when combined with taxol treatment, inducing the cells to apoptosis. These results revealed that the resistance to the anti-proliferative effects induced by taxol can be significantly decreased by combined treatment with selective CRIF1–CDK2 interface inhibitors, that contributes to developing a new treatment that benefits TNBC patients.

## 2. Materials and Methods

### 2.1. Inhibitors and Chemicals

The two CRIF1 inhibitors used in this study were F1142-3225 and F0922-0913, which were obtained as reported [16]. The chemotherapy drug taxol was purchased from Thermo Fisher Scientific (catalog no: P3456, Ontario, ON, Canada). Preparation of stock solutions was performed by dissolving the compounds in dimethyl sulfoxide. Dilutions to the required final concentrations were conducted with culture medium.

### 2.2. Cell Culture

The triple-negative breast cancer cell lines, MDA-MB-231(RRID: CVCL-0062) and MDA-MB-468 (RRID: CVCL-0419), were kindly provided by Dr. Francine Durocher (CHU de Quebec-Research Center and Laval University) on 19 November 2018 (MDA-MB-231 passage number: p + 13, MDA-MB-468 passage number: p + 18). MDA-MB-231 cells at passage number p + 15–p + 21 and MDA-MB-468 cells at passage number p + 20–p + 26 were used in this study. The control cell line MCF10 (CVCL-0598) was a gift from Dr. Charles Doillon (CHU de Quebec-Research Center and Laval University) on 19 November 2015 (passage number p + 2), and cells at passage number p + 4–p + 10 were used in this study. Cells were propagated as previously described [18,19,20]. Fetal bovine serum (FBS) (PAA, Etobicoke, ON, Canada) was added to the medium at a final concentration of 10%. Cells were plated in the medium and, after 24 h, this was replaced by fresh medium containing inhibitors at the required concentrations. Baseline controls (control) were provided by cells cultivated in inhibitor-free medium.

### 2.3. Cell Proliferation Measurement: An Assay for DNA Content with Cell Proliferation Reagent WST-1

The anti-proliferative effects of inhibitors were quantified using the cell proliferation reagent WST-1 (Cat No. 11644807001, Roche Applied Science, Mannheim, Germany). Each cell type was plated into an individual 96-well plate, at a density of 3 × 10^3^ cells per well. The cells underwent a four-day treatment, with inhibitors at the required concentrations. Culture medium was changed every two days. The manufacturer’s instructions were followed throughout the testing procedure. Cell proliferation in the absence of inhibitors was recorded as 100%. DNA synthesis in the inhibitor-treated groups vs. the control group (100%) was compared, and data were reported as a percentage (%). Each condition was assessed five times, and each experiment was conducted three times.

### 2.4. Flow Cytometric Cell Cycle Analysis

Flow cytometry, using the Click-iT EdU assay kit and propidium iodide (PI) (Molecular Probes, Invitrogen, ON, Canada), was used to determine cell cycle distribution. EdU incorporation was monitored as an indicator of DNA synthesis during the S phase. MDA-MB-231, MDA-MB-468, and MCF10A cells were seeded into 6-well plates (2.5 × 10^5^ cells per well). After 24 h, inhibitors at the desired concentrations were added to the cells for an additional 24 h. Cells were harvested, labeled with EdU for 4 h, fixed in 70% ethanol and permeabilized with 0.2% Triton X-100. Propidium iodide was used to label total DNA content. A BD FACS Canto II instrument (BD Bioscience, San Jose, CA, USA) was used for flow cytometric analysis of samples. The percentage of cells, in each of the following phases, were: G0/G1, S, and G2/M were calculated using the Multi-Cycle AV software (Phoenix Flow Systems, San Diego, CA, USA). The percentage of viable cells (G0/G1, S and G2/M = 100%) was reported. Each treatment was assessed three times, and the experiments were conducted three times.

### 2.5. Apoptosis Test

MDA-MB-231 and MCF10A cells were seeded in 6-well culture plates, at a density of 2.5 × 10^5^ cells per well. Twenty-four hours later, 2 μM 3225 or 0913 inhibitor was added to the cells for an additional 24 h. Taxol and inhibitor-treated cells were exposed to 2 μM Taxol, 3225 or 0913, for 24 h. Apoptosis was determined using the eBioscience^TM^ Annexin V Apoptosis Detection kit APC (Catalog no 88880772, Thermo Fisher Scientific, Saint Laurent, QC, Canada), according to the manufacturer’s instructions. Specific procedures: The collected cells were washed with PBS, resuspended in binding buffer, and adjusted to a concentration of 1 × 10^6^ cells/mL. A 100-μL volume of cell suspension was transferred into a flow tube and 5 μL Annexin V-APC was added. After a 10-min. incubation in the dark, the cells were washed with binding buffer. Finally, the cells were suspended in 200 μL binding buffer, and 5 μL PI staining solution was added. This cell suspension was analyzed by the flow cytometry platform service at the CHUL. Each treatment was performed in triplicate, and experiments were repeated three times.

### 2.6. Western Blot

MDA-MB-231 cells were treated with 2 μM 3225 or 0913, in the presence or absence of 2 μM taxol. Total protein extraction was performed with RIPA buffer (Sigma, St. Louis, MO, USA), containing a protease inhibitor cocktail (Calbiochem, Merck, Germany). Protein (40 μg) separation was performed by 12% SDS-polyacrylamide gel electrophoresis. Gels were transferred onto Amersham Hybond P 0.45 PVDF blotting membranes (catalogue no. 10600023, GE Healthcare Life Sciences, Munich, Germany). PBS-Tween 20 containing 5% non-fat milk was used as blocking agent for the PVDS membranes. Blocking was performed for 2 h at room temperature. Primary antibodies were diluted in 5% non-fat milk in PBS-Tween, prior to an overnight incubation at 4 °C with the membranes: anti-CDK2 antibody (ab77671, Abcam, Toronto, ON, Canada), diluted 1:1000. The loading control used was a 1:10,000 dilution of monoclonal anti-β-actin antibody generated in mice (A5316, Sigma, St. Louis, MO, USA). Following washing, the membranes were incubated for 2 h with a goat-anti-mouse IgG-HRP secondary antibody (sc-2005, Santa Cruz Biotechnology, Dallas, TX, USA). Enhanced chemiluminescence (ECL plus Western Blotting Detection Kit, PerkinElmer, Waltham, MA, USA) was used for blot visualization and quantification was carried out using NIH ImageJ software. Relative protein expression level was calculated using the ratio obtained between the protein of interest and β-actin. Three independent trials of each experiment were conducted.

### 2.7. Cytotoxicity Analysis

The 3 × 10^3^ cells were seeded into 96-well plates, with 100 μL culture medium, and grown for 24 h. The cells were then treated for 4 days with inhibitors at final concentrations of 2.5, 5, 10, 20, 40, 80, and 160 μM. Cell viability was determined by WST-8 reagent, according to the method described above. Untreated cells were used as the control. Each condition was conducted in triplicate, and the experiments were repeated three times.

### 2.8. CDK2 Knockdown

CDK2 knockdown was conducted using small interfering RNA (siRNA) transfection. SiRNA oligos against human CDK2 and non-targeting siRNA were synthesized by oligobio (Appendix A). At a density of 3000 cells per well, the two TNBC cell-lines, MDA-MB-231 and MDA-MB-468, and the control cell-line, MCF-10A, were transfected with 175 nM CDK2 siRNA or 100 nM NT siRNA using the Lipofectamine 2000 (Invitrogen, Burlington, ON, Canada), according to the manufacturer’s instructions. After 5 h of treatment with siRNA, the medium was changed, and 2 nM of taxol was added to each treatment group. For each cell-line, the control group was taxol and siRNA free. Experiments were repeated three times for each cell line. Cell medium was changed after 48 h, and the cell proliferation test was conducted after 4 days of treatment.

### 2.9. Statistical Analysis

Two groups were compared using the Student’s *t*-test, whereas ANOVA was used for multiple comparisons. Data were presented as means ± SD. Statistically significant differences were determined by *p* ˂ 0.05, *p* ˂ 0.01, and *p* ˂ 0.001.

## 3. Results

### 3.1. The CRIF1–CDK2 Interface Inhibitors Promoted the Transition of the Cell from G0/G1 to S Phase in TNBC Cell Lines

In order to circumvent the non-specificity problem, encountered in earlier attempts to target CDKs, our study indirectly targets CDK2 in TNBC by taking advantage of the cell cycle regulator CRIF1, through which we expected to overcome the resistance to chemotherapy in TNBC. Two previously reported CRIF1–CDK2 interface inhibitors, F1142-3225 and F0922-0913, were used to see if they were effective [16]. The cell proliferation reagent WST-1 assay was used to study the role of inhibitors on two typical TNBC cell lines, MDA-MB-231 and MDA-MB-468. In addition, the human mammary epithelial cell line MCF10A was used as a model for a normal breast cell line [21,22]. As shown in Figure 1A, a dose-dependent inhibition of proliferation was observed, with IC_50_ s of 49.9 ± 2.7 μM and 33.4 ± 3.6 μM for F1142-3225 and F0922-0913, respectively, in MDA-MB-231 cells. Similar trends were observed for MDA-MB-468 cells, with IC_50_ values of 39.5 ± 3.4 μM and 27.0 ± 3.3 μM for F1142-3225 and F0922-0913, respectively (Figure 1B).

Failure of the cell cycle check-point arrest responses can be taken advantage of for therapeutic use. Disruption of the checkpoint control in cells increases their sensitivity to further microtubular or genotoxic damage [23]. Therefore, the cell cycle checkpoint regulation, induced by F1142-3225 and F0922-0913, could potentially be used to increase the cell sensitivity to chemotherapy. Results indicated that 2 μM F1142-3225 and F0922-0913 induced significant cell cycle modulation in MDA-MB-231 and MDA-MB-468 cells. MDA-MB-231 cells in S phase increased from 8.5 to 13.4% (Figure 2A). The percentage of MDA-MB-468 cells in S phase increased from 8.3 to 14.6% (Figure 2B). These interface inhibitors can demonstrate a very significant role at relative low concentration, compared to their IC_50_, as was also the case in the study of Ran et al. [16].

Meanwhile, the percentage of cells in the G0/G1 phase decreased, from approximately 78.4 to 69.3%, in both TNBC cell lines. There was no significant change in the percentage of cells in the G2M phase, which was approximately 13.6% in both cell lines. In contrast, the cell cycle in MCF10A cells was not modulated by the two CRIF1–CDK2 interface inhibitors (Figure 3A). Furthermore, it was also found that the CDK2 protein level was up-regulated (Figure 3B).

The above results indicate that these two CRIF1–CDK2 interface inhibitors led to the increase in CDK2 protein level in the absence of taxol, advancing cell transition from G0/G1 to S phase. Effect of F1142-3225 or F0922-0913 on CRIF1 expression is shown in Appendix A. 

#### 3.1.1. TNBC Cell Cycle Regulation

When combined with taxol treatment, the CRIF1–CDK2 interface inhibitors could advance the Cells from the G0/G1 to S phase and G2/M phase in TNBC cell lines

Taxol is a microtubule-stabilizing drug that has approval from the Food and Drug Administration to treat breast cancer. It functions by blocking cells in the G2/M phase of the cell cycle, thereby stopping cells from being able to form a normal mitotic apparatus, ultimately resulting in cell death [24]. During this process, microtubule dynamics have an important role in the mitotic checkpoint and chromosome movement, and their blockade is what permits taxol to inhibit mitotic progression and cell proliferation [25]. Furthermore, it is reported that the effective concentration of taxol, to suppress microtubule dynamics in Caov-3 ovarian adenocarcinoma cells and A-498 kidney carcinoma cells, lies between 30 nM to 100 nM [25]. Recent evidence points to intratumoral concentrations of taxol being too low to cause mitotic arrest [24], and this might be related to the cells’ resistance to taxol chemotherapy.

In our study, we verified that the TNBC cells exhibited resistance to taxol, with a concentration higher than 18 nM (Appendix A). In order to limit taxol toxicity, a concentration of 2 nM was selected for this study. MDA-MB-231 and MDA-MB-468 cells were treated with 2 nM taxol, in combination with either F1142-3225 or F0922-0913. Cell proliferation in MDA-MB-231 was decreased by 20% (to 80%) by taxol. However, it was further reduced by 30 (to 50%) and 36% (to 44%), in response to the addition of 4 μM F1142-3225 and F0922-0913, respectively (Figure 4A). Similar trends were observed in MDA-MB-468 cells, where combination treatment of 4 μM F1142-3225 or F0922-0913 and taxol could further decrease proliferation to approximately 44%, which is markedly more significant than the proliferation reduction induced by taxol alone (18.8% reduction) (Figure 4B).

Furthermore, the effect of inhibitors with taxol treatment was also determined in MCF10A cells. It was shown that there was no significant change in proliferation in response to the combination treatment of F1142-3225 or F0922-0913 with taxol (approximately 77.9% vs. 80.0%) (Figure 5). Thus, it was demonstrated that the combined administration of F1142-3225 or F0922-0913 could significantly increase the cells’ sensitivity to taxol in the TNBC cell lines, but not in MCF10A cells.

Meanwhile, the cell cycle distribution was analyzed, following the combination treatment of either of these two CRIF1–CDK2 interface inhibitors with taxol. In MDA-MB-231, the percentage of cells in the G2/M phase was increased, in response to taxol treatment (13.2 vs.18.6%). This trend was significantly enhanced by the combined use of taxol with F1142-3225 or F0922-0913 (18.6 vs. 27.4%) (Figure 6A). In MDA-MB-468, the percentage of cells in G2/M phase, following taxol treatment, was 19.9%, and this was increased to approximately 29.2%, in response to the combined treatment with F1142-3225 or F0922-0913 (Figure 6B). It is evident that combined treatment resulted more significant S and G2/M phase increase (compared Figure 6 with Figure 2).

In contrast, in MCF10 cells, there was no significant regulation of G2/M arrest in groups of combination treatments, compared with the cells just treated with taxol (23.6% vs. 23.4%) (Figure 7A). Furthermore, following combination treatment of taxol and the interface inhibitors, CDK2 protein expression was significantly decreased to 59.9%, compared with the group treated with taxol alone (Figure 7B). Thus, it is suggested that the down-regulation of CDK2 in the combined treatment can arrest the TNBC cancer cell cycle and enhance the anti-proliferation effects, imposing a positive effect on the treatments for breast cancer [26,27,28].

#### 3.1.2. Effect of CDK2 Knock Down on TNBC Cell Proliferation

Effect of CDK2 knockdown in chemosensitization, estimated by a cell proliferation assay, is shown for MDA-MB-231 and MDA-MB-468 cells (Figure 8). TNBC cells with CDK2 knockdown showed significantly increased sensitivity to taxol treatment. In both TNBC cells, CDK2 knockdown led to more decrease in cell proliferation in the presence of taxol (2 nM) than in the absence. This could be related to changes induced by the CDK2 regulatory phosphorylation [16]. The chemosensitization in CDK2-silenced cells reflect a similar behavior to the CRIF1–CDK2 interface inhibitors, supporting the role of these inhibitors in disrupting interactions between these two molecules.

### 3.2. The CRIF1–CDK2 Interface Inhibitors Induced Apoptosis in TNBC Cell Line but Not in MCF10A

It has been reported that, although the toxicity of chemotherapeutic agents, such as taxol, focus on the cell cycle, cancer cells may adopt various DNA repair mechanisms or depend on impaired checkpoints to reverse or avoid drug-induced damage to DNA [28]. However, the CRIF1–CDK2 interface inhibitors advance cells from the G0/G1 to S phase and G2/M, producing irreparable cells that undergo apoptosis, as observed in Figure 9. Cellular apoptosis, induced by treatment with a combination of the inhibitors and taxol, was studied in the MDA-MB-231 and MCF10A cell lines. As shown in Figure 9A, apoptosis was significantly elevated, in response to application a combination of inhibitors and taxol, whereas MCF10A cells were not affected (Figure 9B). These differences could be attributed to frequent tumor cell dependence on individual CDKs and cyclins, suggesting that inhibitors could selectively target cancer cells [29].

### 3.3. No Significant Cytotoxicity Was Observed for the CRIF1–CDK2 Interface Inhibitors within the Treatment Concentration in MCF10A Cell Line

A major concern with a CDK-targeting agent is toxicity: low selectivity underlies the primary cause of failure in the majority of clinical trials [17,19]. The cytotoxicity of the two inhibitors was studied in MCF10A cells using the Cell Proliferation Reagent WST-1 kit. No significant toxicity was observed up to 160 μM for the inhibitors, suggesting a significant therapeutic window for these drugs (Appendix A).

## 4. Discussion

Chemotherapy is the standard treatment for TNBC. However, more than 50% of patients show resistance to chemotherapy. In our study, we demonstrated a resistance of TNBC cells to taxol, with a concentration higher than 18 nM (Appendix A). However, 2 μM F1142-3225 or F0922-0913 combined 2 nM taxol can decrease the TNBC cell proliferation by 50% (to 50%), which is much higher than the 20% reduction (to 80%), induced by 2 nM taxol (Figure 4). Meanwhile, the cell cycle revealed that there was a significant arrest for TNBC cells in the G2/M phase and progress of cells in apoptosis, which was induced by the joint treatment of CRIF1–CDK2 interface inhibitors and taxol (Figure 6). This indicated that these interface inhibitors are affecting the CRIF1–CDK2 complex, significantly activating the G1/S checkpoint and favoring increased cell distribution in the G2/M phase, when administered with taxol.

In a previous study, after ionizing radiation, it was shown that the CRIF1 promotes CDK2 nuclear translocation and phosphorylation on T14/T160, facilitating G1/S checkpoint activation and DNA damage repair. The knockdown of CRIF1 was shown to block the nuclear translocation of CDK2, which results in DNA damage repair delay and increased radiosensitization [20]. It is theoretically possible that these two inhibitors blocked the interaction of CRIF1 and CDK2, promoting cell transition from G0/G1 to the S phase. Taxol treatment arrested the cells at the G2/M checkpoint, followed by induction of the damage repair response. Our result made it clear that the action of CDK2 interface inhibitors and knockdown have similar roles in the presence of taxol on TNB cells. It is reported that attempts to promote G2/M arrest have also correlated with enhanced apoptosis [30]. Therefore, with the arrest of G2/M checkpoint by CRIF1–CDK2 interface inhibitors, the cell apoptosis of TNBC cell lines were significantly up-regulated. Based on our previous study in OS cells, the elevated activity of CDK2, demonstrated by T160 phosphorylation and the S807 and S811 phosphorylation of Rb, may have promoted the G2/M arrest of cells [16].

On the other hand, there was no significant regulation of cell cycle and apoptosis in the normal cell model MCF10A by the CRIF1–CDK2 interface inhibitors (Figure 7A), which demonstrated that these inhibitors have a high specificity to TNBC cells. Normal CDKs inhibitors target CDKs, such as CDK4/6, directly to regulate the cell cycle, decreasing the cell proliferation by arresting cells in the G0/G1 or G2/M checkpoints [11,12,31]. However, numerous CDK inhibitors have indicated low specificity and selectivity to target-specific CDK enzymes [3,17]. The CRIF1–CDK2 interface inhibitors show benefits to the specificity and selectivity of treatment because they target CRIF1 and CDK interfaces, thus indirectly modulating the cell cycle and apoptosis and increasing the cancer cell sensitivity to chemotherapy.

## 5. Conclusions

Therefore, combination treatment of the CRIF1–CDK2 interface inhibitors and taxol has the potential to kill TNBC cells and enhance cell sensitivity by promoting G2/M arrests and apoptosis. Moreover, this cell cycle regulation is selective to TNBC cells, with no effect on normal cells.

## Figures and Tables

**Figure 1 cancers-14-00989-f001:**
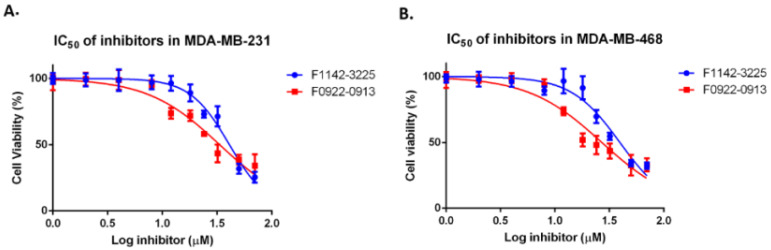
The IC50 plots of F1142-3225 and F0922-0913 on cell proliferation of MDA-MB-231 and MDA-MB-468. Cell proliferation after treatment of MDA-MB-231 (**A**) or MDA-MB-468 cells (**B**), with F1142-3225 or F0922-0913 for 4 days. Cell proliferation was determined by cell proliferation reagent WST-1. Data are reported as the mean ± SD (*n* = 3) of the individual experiments. The experiment was independently repeated three times.

**Figure 2 cancers-14-00989-f002:**
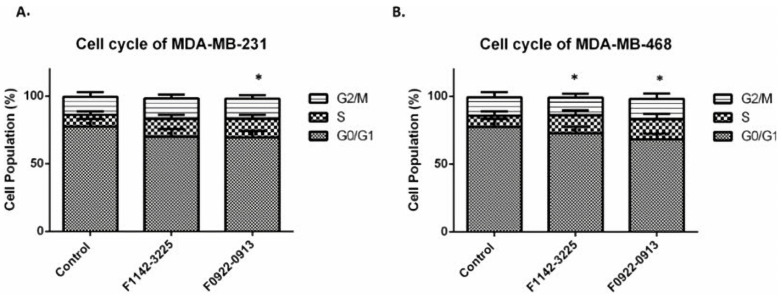
The cell cycle distribution of MDA-MB-231 and MDA-MB-468 cells in response to inhibitors. (**A**,**C**) Cell cycle distribution modification, induced by F1142-3225 or F0922-0913 in MDA-MB-231 cells; (**B**,**D**) cell cycle distribution, induced by F1142-3225 or F0922-0913 in MDA-MB-468 cells. Data are presented as % of living cells (G0/G1, S, and G2/M cells = 100%). Each value represents the mean of experiments carried out in triplicate (mean ± SD). Statistical significance by one-way ANOVA; * *p* ˂ 0.05 vs. control (ctrl). Flow cytometry figures depict the cellular distribution in each phase of the cell cycle after treatment.

**Figure 3 cancers-14-00989-f003:**
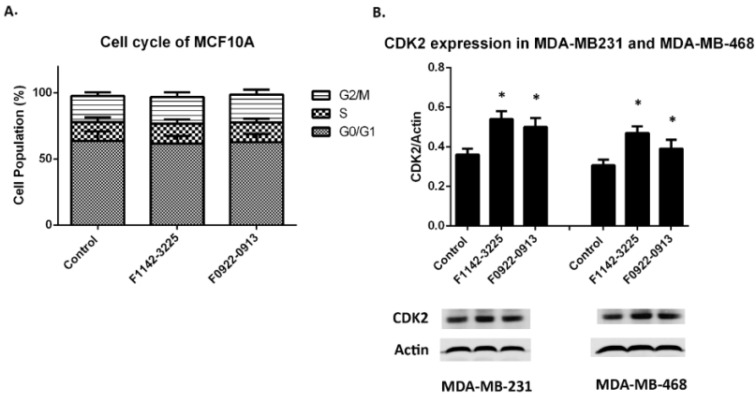
The cell cycle distribution of MCF10A cells and CDK2 protein expression in TNBC cell lines. (**A**) Cell cycle distribution in response to F1142-3225 or F0922-0913 in MCF10A cells. Data are presented as % of living cells (G0/G1, S, and G2/M cells = 100%). Each value represents the mean of experiments, carried out in triplicate (mean ± SD). Statistical significance was determined by one-way ANOVA. (**B**) CDK2 protein expression after treatment with F1142-3225 or F0922-0913, for 48 h, was determined by western blot in MDA-MB-231 and MDA-MB-468 cells. Data are reported as the mean ± SD (*n* = 3) of the individual experiments. Statistical significance was determined by Student’s test: * *p* < 0.05 vs. control (ctrl). Detailed information about Western Blot can be found at Appendix A.

**Figure 4 cancers-14-00989-f004:**
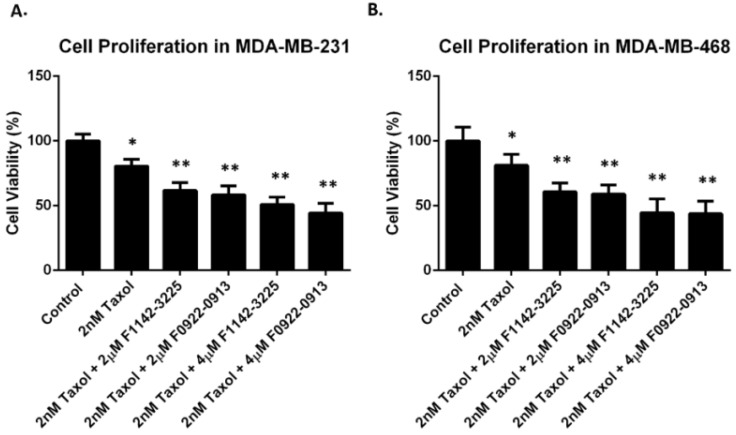
Cell proliferation in response to the combined treatment of taxol and inhibitors in TNBC cell lines. Cell proliferation determined by cell proliferation reagent WST-1 in MDA-MB-231 (**A**) and MDA-MB-468 (**B**) cells treated with taxol, in the presence or absence of F2114-3225 or F0922-0913, for 4 days. Data are reported as % of DNA synthesis vs. control (100%). Each point represents the mean of four replications (mean ± SD). Statistical significance was determined by one-way ANOVA; * *p* ˂ 0.05 vs. control (ctrl); ** *p* ˂ 0.01 vs. control (ctrl).

**Figure 5 cancers-14-00989-f005:**
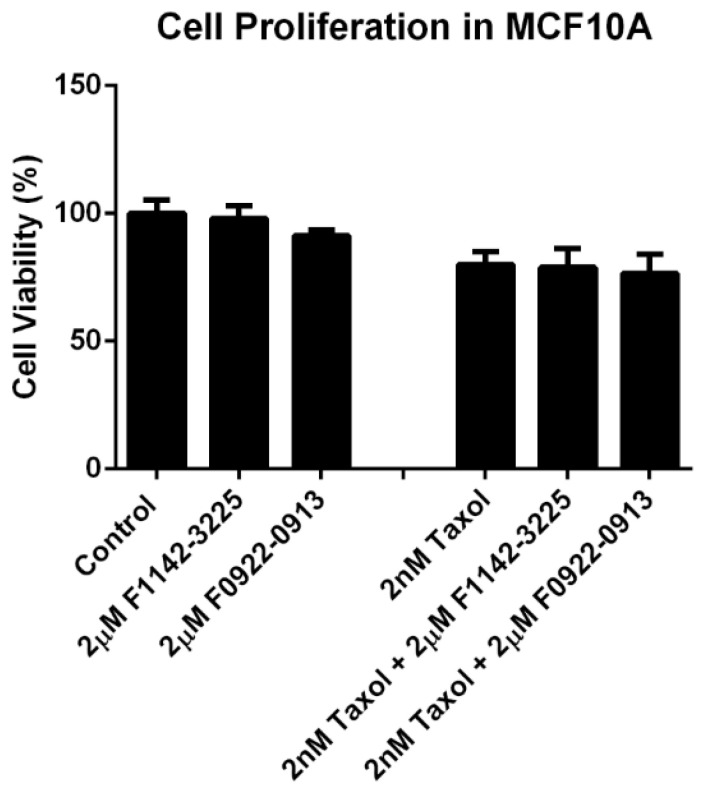
The cell proliferation in the presence of inhibitors with or without taxol in the MCF10A cell line. Cell proliferation was determined by cell proliferation reagent WST-1 in MCF10A cells treated with F1142-3225 or F0922-0913, in the presence or absence of taxol, for 4 days. Data are reported as % of DNA synthesis vs. control (100%). Each point represents the mean of four replications (mean ± SD). Statistical significance by one-way ANOVA.

**Figure 6 cancers-14-00989-f006:**
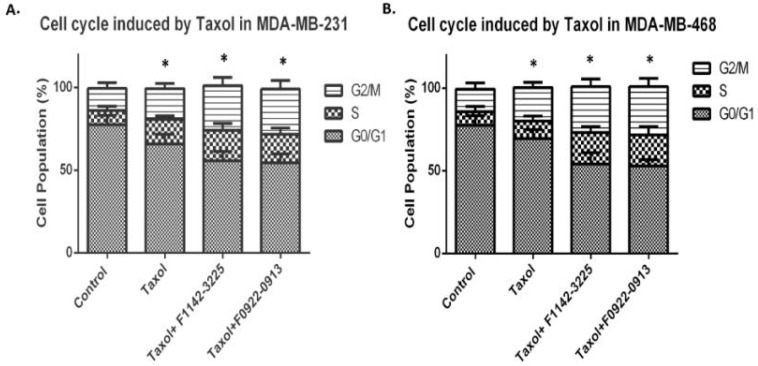
Cell cycle distribution in response to combined treatment of taxol and inhibitors in MDA-MB-231 and MDA-MB-468 cells. (**A**,**C**) Cell cycle distribution in response to combination treatment of taxol and either F1142-3225 or F0922-0913 in MDA-MB-231 cells. (**B**,**D**) Cell cycle distribution in response to combination treatment of taxol and either F1142-3225 or F0922-0913 in MDA-MB-468 cells. Data are presented as % of living cells (G0/G1, S, and G2/M cells = 100%). Each number represents the mean of experiments, carried out in triplicate (mean ± SD). Statistical sig-nificance was determined by one-way ANOVA; * *p* ˂ 0.05 vs. control (Ctrl). Flow cytometry figures depict the distribution of each phase of the cell cycle after treatment.

**Figure 7 cancers-14-00989-f007:**
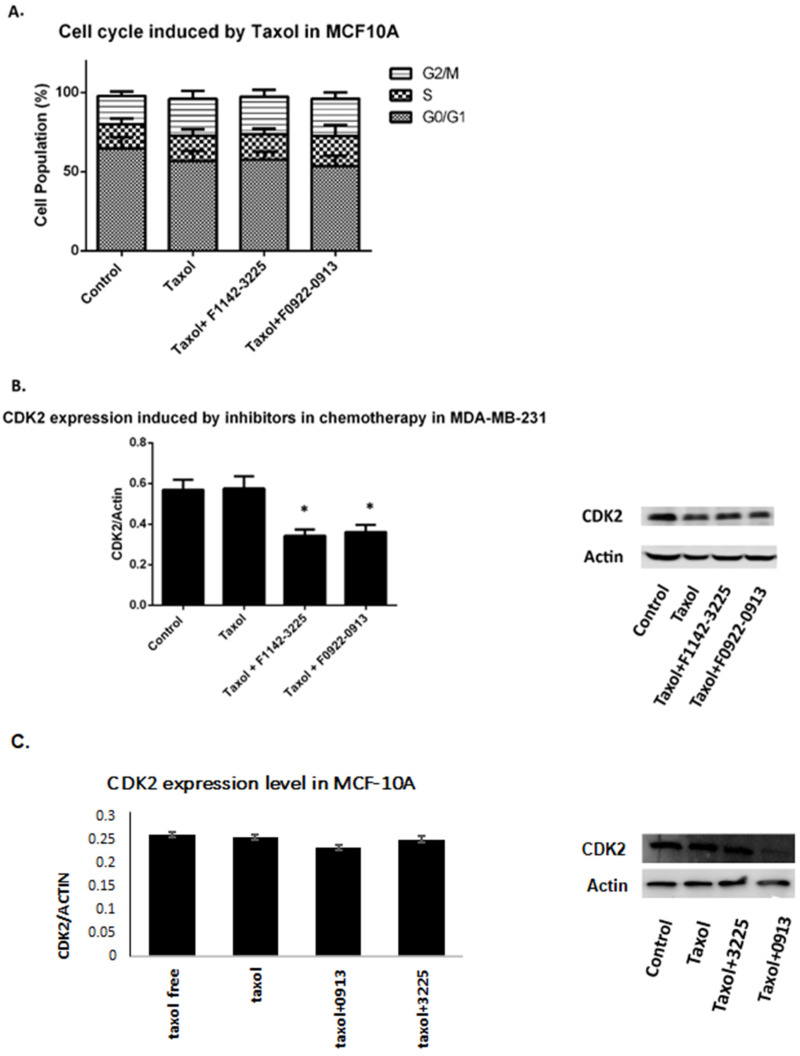
The cell cycle distribution of MCF10A cells and CDK2 protein expression in response to combination treatment in TNBC cell lines. (**A**) Cell cycle distribution in response to combination treatment of taxol with either F1142-3225 or F0922-0913 in MCF10A cells. Data are presented as % of living cells (G0/G1, S, and G2/M cells = 100%). Each number represents the mean of experiments, carried out in triplicate (mean ± SD). Statistical significance was determined by one-way ANOVA. (**B**) CDK2 protein expression, following combination treatment of taxol with either F1142-3225 or F0922-0913 for 48 h, was determined by western blot in MDA-MB-231 cells and MDA-MB-468 cells. (**C**) CDK2 protein expression in the control cell line MCF-10A, following combination treatment of taxol with either F1142-3225 or F0922-0913 for 48 h, was determined by western blot. Data are reported as the mean ± SD (*n* = 3) of the individual experiments. Statistical significance was determined by Student’s test: * *p* < 0.05 vs. control (Ctrl). The above results indicate that CRIF1–CDK2 interface inhibitor F1142-3225 has no significant affect on the CDK2 protein expression level in normal cells. Detailed information about Western Blot can be found at Appendix A.

**Figure 8 cancers-14-00989-f008:**
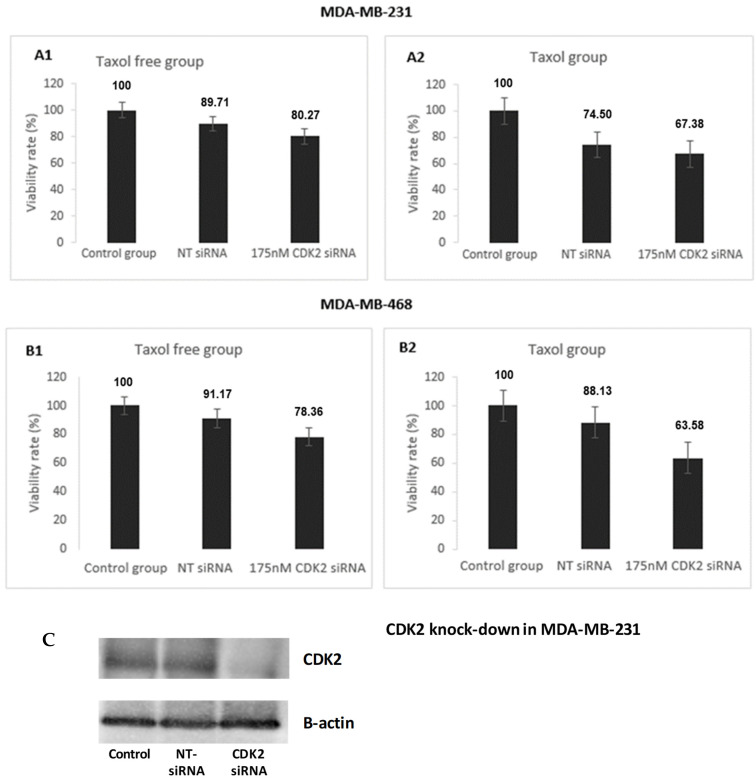
Effect of CDK2 knockdown on the proliferation of MDA-MB-231 and MDA-MB-468. (**A1**) Taxol-free group in MDA-MB-231. (**A2**) Taxol group in MDA-MB-468. (**B1**) Taxol-free group in MDA-MB-468. (**B2**) Taxol group in MDA-MB-468. (**C**) Confirmation that the CDK2 has been successfully knocked down in TNBC cells.

**Figure 9 cancers-14-00989-f009:**
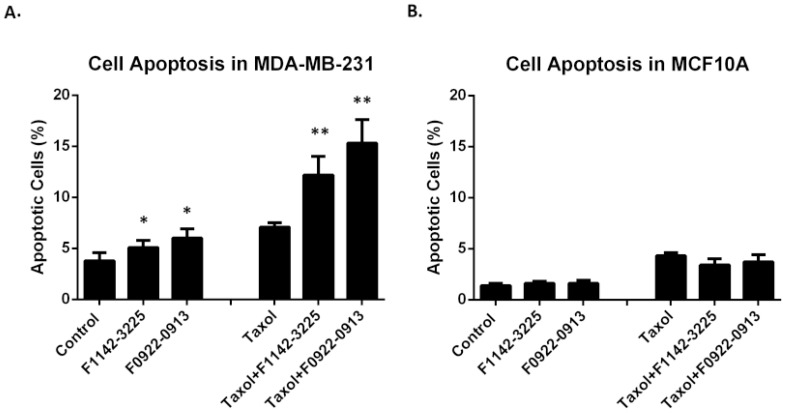
Cell apoptosis induced by the inhibitors with or without taxol in TNBC and MCF10 A cell lines. Cell apoptosis was determined by eBioscience^TM^ Annexin V Apoptosis Detection kit APC in MDA-MB-231(**A**) and MCF10A (**B**) cell lines treated with either F2114-3225 or F0922-0913, in the presence or absence of taxol for 24 h. Data are reported as % of DNA synthesis vs. control (100%). Each point represents the mean of four replications (mean ± SD). Statistical significance by one-way ANOVA; * *p* ˂ 0.05 vs. control (ctrl); ** *p* ˂ 0.01 vs. control (ctrl).

## Data Availability

The data presented in this study are available on request from the corresponding author.

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
