# Peer review of "CRIF1-CDK2 Interface Inhibitors Enhance Taxol Inhibition of the Lethal Triple-Negative Breast Cancer"

_cancers, 2022, doi:10.3390/cancers14040989_

Round 1

Reviewer 1 Report

The authors have addressed most of my prior comments. However,  the authors only show the expression levels of CDK2, it is still not clear about the CDK2 activity, which is more important. 

Author Response

Please find the specific reply (including figure) in the attachment.

Reviewer 1: The authors have addressed most of my prior comments. However,  the authors only show the expression levels of CDK2, it is still not clear about the CDK2 activity, which is more important. 

The CDK2 activity was studied by us using the same interface inhibitors to CDK2/CRIF1 complex in U2OS and SaOS2 cancer cells and reported in our previous publication (Ran et al, Journal of American Chemical Society, 2019, 41:1420). Replicating the same study in TNB cell is absolutely possible but with current significant delay in supply chain in the pandemics, it will surely take more than 2 months to receive the raw materials for such studies. This is not convincing to the time scale given by the editorial office. As the CDK2 activity was unambiguously demonstrated with the same inhibitors to CDK2/CRIF1 interaction in our previous publication, we have now indicated this point in the discussion of the new manuscript version.  In this paper, we mainly demonstrated that the interface inhibitors are also applicable in TNBC cells in the combined treatment with Taxol, resulting in advancing the cells from G0/G1 to S and G2/M phases, producing irreparable cell damage which then undergoes apoptosis.

In our previous publication, we have studied specific phosphorylation rate in retinoblastoma (Rb), the substrate of CDK2 that demonstrated its activity. T160 phosphorylation of CDK2 and the S807 and S811 phosphorylation of Rb were significantly elevated in inhibitor treated cells in comparison with the control group (figure shown in the attachment).

Phosphorylation analysis of Rb in response to the drug molecule intervention, before (IR-) and after (IR+) the radiation treatment. β-actin is shown as a loading control. (cited from Figure S6, Ran et al. 2019).

Reviewer 2 Report

I found that the revision was performed in a sound manner scientifically.

Author Response

Thanks for your comments and suggestions

Reviewer 3 Report

concerns addressed

Author Response

(The authors gave the same response as above.)

Round 2

Reviewer 1 Report

While I agree with the author about the potential challenge. However, the experiment should have been done in the first place as small molecule modulators do not always behave exactly the same in different cells. The current discussion "Elevated activity of CDK2 406 demonstrated by T160 phosphorylation and the S807 and S811 phosphorylation of Rb may 407 have promoted the G2/M arrest of cells[16]" sounds like the phosphorylation status have been demonstrated, which is confusing. So the authors should clearly state that this is not the case, but based on the previous publication, it may contribute to the G2/M arrest.

Author Response

  1. While I agree with the author about the potential challenge. However, the experiment should have been done in the first place as small molecule modulators do not always behave exactly the same in different cells.

We agree that it should be better that we had first studied the CDK2 activity in TNB cells. At that time, we simply did experiments following the comments, without knowing the supply chain problem can be so important.

  1. The current discussion "Elevated activity of CDK2 406 demonstrated by T160 phosphorylation and the S807 and S811 phosphorylation of Rb may 407 have promoted the G2/M arrest of cells[16]" sounds like the phosphorylation status have been demonstrated, which is confusing. So the authors should clearly state that this is not the case, but based on the previous publication, it may contribute to the G2/M arrest.

We agree with the comment that the last discussion about elevated activity was confusing. We have now modified the discussion into: ‘Based on our previous study in OS cells,  the elevated activity of CDK2 demonstrated by T160 phosphorylation and the S807 and S811 phosphorylation of Rb may have promoted the G2/M arrest of cells [16].

This manuscript is a resubmission of an earlier submission. The following is a list of the peer review reports and author responses from that submission.

Round 1

Reviewer 1 Report

This manuscript by Sang and Lin titled “CRIF1-CDK2 interface inhibitors enhance Taxol inhibition of the lethal triple-negative breast cancer” reports that CRIF1-CDK2 interface inhibitors led to the increase in CDK2 protein level, advancing cell transition from G0/G1 to S phase, and sensitizing TNB cells to Taxol induced cell proliferation inhibition and apoptosis. While these findings are interesting, the manuscript in its current form is a bit too preliminary for publication.

Figure 3B: The authors showed that F1142-3225 or F0922-0913 treatment increased CDK2 protein levels in TNB cells. Control CDK2 levels in MCF10A cells should also be shown. Also, what about CDK2 activity, which is more interesting than the protein level itself?

The notion that increased cellular CDK2 activity sensitizing chemo-sensitivity in TNB cells is counterintuitive and not convincing based on the data presented. The authors should at least validate the hypothesis by testing the chemo-sensitivity of TNB cells with CDK2 knocked down or with CDK2 overexpression.  

In addition to Taxol, have the authors tested other TNB chemotherapeutic agents such as capecitabine?

Author Response

Reviewer 1:

This manuscript by Sang and Lin titled “CRIF1-CDK2 interface inhibitors enhance Taxol inhibition of the lethal triple-negative breast cancer” reports that CRIF1-CDK2 interface inhibitors led to the increase in CDK2 protein level, advancing cell transition from G0/G1 to S phase, and sensitizing TNB cells to Taxol induced cell proliferation inhibition and apoptosis. While these findings are interesting, the manuscript in its current form is a bit too preliminary for publication.

  • Figure 3B: The authors showed that F1142-3225 or F0922-0913 treatment increased CDK2 protein levels in TNB cells. Control CDK2 levels in MCF10A cells should also be shown. Also, what about CDK2 activity, which is more interesting than the protein level itself?

Yes, we showed that these interface inhibitors increased CDK2 protein levels in the absence of Taxol, in TNB cells, shown by Fig. 3B in the last manuscript version. But in the presence of Taxol, the inhibitor effect led to the reduction of CDK2 expression, shown by Fig. 7B in the last manuscript version. Following your comment, we carried out the same experiment in MCF10-A in the presence of Taxol, from which we saw no significant CDK2 expression reduction in the co-presence of F1142-3225 or F0922-0913 as compared to the control with Taxol (99% and 93% respectively, within experimental errors). This demonstrates the inhibitors are specific toward TNB cells. The results in MCF10A cells appears now as Fig. 7C in our revised manuscript.

  • The notion that increased cellular CDK2 activity sensitizing chemo-sensitivity in TNB cells is counterintuitive and not convincing based on the data presented. The authors should at least validate the hypothesis by testing the chemo-sensitivity of TNB cells with CDK2 knocked down or with CDK2 overexpression.

Thank you very much for giving insight that we carried out series of experiments, which has significantly elevated the quality of the manuscript. Following the comment, we have knocked down CDK2 in TNB cells (MDA-MB-231 and -468) and found that the viability of the cells decreased in the absence of Taxol, and such decrease is even more significant in the presence of Taxol (Fig. 8). This is in coincidence to the case of CDK2 expression in the presence of Taxol and interface inhibitors, in which CDK2 expression is decreased significantly (Fig.7, A and B). So the notion that increased cellular CDK2 activity sensitizing chemo-sensitivity in TNB cells should be corrected.

In an earlier study, a combination of either CDK2 or EZH2 inhibitor with tamoxifen also effectively suppressed tumor growth and markedly improved the survival of the mice bearing TNBC tumors (https://www.nature.com/articles/s41467-019-13105-5). Mechanism of radiation sensitization by inhibiting CRIF1 inhibition was reported in a previous study(https://www.pnas.org/content/116/41/20511.short)

At this point, we agree with the comment of the reviewer, that  a brief statement of CDK2 activity can not well explain the results, thus  modifications have been carried out in the manuscript,  in the text around Fig. 3 and 7. In our case, it is now clear that CDK2 knock down and interface inhibitors have similar roles in the presence of Taxol on TNB cells.

  • In addition to Taxol, have the authors tested other TNB chemotherapeutic agents such as capecitabine?

We focused on Taxol, which is the most used chemotherapy. In the pandemic period, the supply chain has really problems.

Reviewer 2 Report

The authors demonstrated that the combination treatment of the CRIF1-CDK2 interface inhibitors and Taxol has the potential to kill TNBC cells and enhance cell sensitivity by the promotion of G2/M arrests and apoptosis. Moreover, this cell cycle regulation is selective to TNBC cells with no effect on normal cells. It could be helpful to TNBC patients.

  1. 2.8. Statistical analysis: corrections for multiple comparisons using Bonferroni, Holm or by controlling the false discovery rate (FDR) were performed?
  2. Discussion about CRIF1-CDK2 interface inhibitors for other type of cancers is recommended.

Author Response

Reviewer 2:

Comments and Suggestions for Authors

The authors demonstrated that the combination treatment of the CRIF1-CDK2 interface inhibitors and Taxol has the potential to kill TNBC cells and enhance cell sensitivity by the promotion of G2/M arrests and apoptosis. Moreover, this cell cycle regulation is selective to TNBC cells with no effect on normal cells. It could be helpful to TNBC patients.

Reviewer 3 Report

Summary: Reconsider after major revision.

The authors demonstrate a potential combination of Taxol and CRIF-CDK inhibitors to increase sensitivity to Taxol. The idea is intriguing and relevant to Cancers, but there is no in vivo demonstration and lacks some key in vitro experiments as well.

  • Despite presenting a clinically relevant idea, there is no in vivo demonstration of the synergy between Taxol and Crif1 inhibitors. In vivo efficacy is needed to establish if this combination is a valid strategy in TNBC.
  • Resistance to Taxol is mentioned several times in the manuscript, but all experiments are performed on regular cell lines and not their Taxol resistant counterparts (such as MDA-MB-231/PacR). This dampens the claims made in the manuscript.
  • What was the criteria for establishing that cells were resistant to Taxol at a concentration higher than 18 nM? The authors claim that the cell lines were resistant to 18 nM Taxol, but cell viability was less than 50% at these concentrations (Fig. S1). The IC50 of MDA-MB-231 cells to Taxol seems to be around 4-8 nM, which is standard for these cells.
  • What is the level of expression of CDK2 in MCF10A cells? This band should be present alongside the TNBC cell lines in Fig. 3B
  • What is the level of CRIF1 expression in MCF10A and the TNBC cell lines? CRIF1 bands are needed in Fig. 3B and 7B.
  • Is the quantification of CDK2 levels in Fig. 7B reflective of the blot on the right? If so, why does the CDK2 expression go down in the blot but not in the quantification? Also, combination of Taxol and CRIF-CDK inhibitors (3rd column) seems to increase CDK2 levels compared to Taxol alone (2nd column), which is consistent with Fig. 3B but is not shown in the quantification.
  • Please plot Fig. S2 the same way as Fig. 1

Author Response

Reviewer 3:

  • 8. Statistical analysis: corrections for multiple comparisons using Bonferroni, Holm or by controlling the false discovery rate (FDR) were performed?

We checked the necessity for use these corrections (https://onlinelibrary.wiley.com/doi/10.1111/opo.12131), such as a fairly small number of multiple comparisons may need Bonferroni correction (when a single false positive in a set of tests would be a problem). We have not found it necessary for the present study.

  • Discussion about CRIF1-CDK2 interface inhibitors for other type of cancers is recommended. The authors demonstrate a potential combination of Taxol and CRIF-CDK inhibitors to increase sensitivity to Taxol. The idea is intriguing and relevant to Cancers, but there is no in vivo demonstration and lacks some key in vitro experiments as well.

 Despite presenting a clinically relevant idea, there is no in vivo demonstration of the synergy between Taxol and Crif1 inhibitors. In vivo efficacy is needed to establish if this combination is a valid strategy in TNBC.

CRIF1-CDK2 interface inhibitors for other type of cancers can be for osteosarcoma (Ran et al.,  J. Am. Chem. Soc. 2019, 141, 1420−1424).

We need to publish the in vitro paper followed by project application to be able to carry out in vivo assay.

 3)   Resistance to Taxol is mentioned several times in the manuscript, but all experiments are performed on regular cell lines and not their Taxol resistant counterparts (such as MDA-MB-231/PacR). This dampens the claims made in the manuscript.

Information for MDA-MB-231/PacR cell-line:

We have not been able to perform the same experiences using this cell-line because of its non- availability from ATCC and similar company. We also contacted the original laboratory elaborating these cells belonging to the Department of Pharmacology & Toxicology, Indiana University, we have not received any response from them.

Article reference for the MDA-MB-231/PacR cell-line: (Sprouse AA, Herbert BS. Resveratrol augments paclitaxel treatment in MDA-MB-231 and paclitaxel-resistant MDA-MB-231 breast cancer cells. Anticancer Res. 2014 Oct;34(10):5363-74. PMID: 25275030).

  • What was the criteria for establishing that cells were resistant to Taxol at a concentration higher than 18 nM? The authors claim that the cell lines were resistant to 18 nM Taxol, but cell viability was less than 50% at these concentrations (Fig. S1). The IC50 of MDA-MB-231 cells to Taxol seems to be around 4-8 nM, which is standard for these cells.

This is because that the cell viability does not further decrease from 18 nM Taxol, with increasing  Taxol concentration.

  • What is the level of expression of CDK2 in MCF10A cells? This band should be present alongside the TNBC cell lines in Fig. 3

CDK2 expression is now studied in MCF10A cells and shown in the new Fig.7C.  Before addition of inhibitors, CDK2/Actin is about 0.25 in Taxol free MCF10-A cells, while CDK2/Actin Is close to 0.38 in MDA-MB-231 cells, and 0.32 in MDA-MB-468 cells.

  • What is the level of CRIF1 expression in MCF10A and the TNBC cell lines? CRIF1 bands are needed in Fig. 3B and 7B.

CRIF1/Actin is about 0.8  in MCF10-A cells (absence of Taxol and inhibitors), while CRIF1/Actin is about 0.67 in MDA-MB-231 cells, and 0.85 in MDA-MB-468 cells (Fg. S3).  

  • Is the quantification of CDK2 levels in Fig. 7B reflective of the blot on the right?,,, If so, why does the CDK2 expression go down in the blot but not in the quantification? Also, combination of Taxol and CRIF-CDK inhibitors (3rd column) seems to increase CDK2 levels compared to Taxol alone (2nd column), which is consistent with Fig. 3B but is not shown in the quantification.

The quantification of CDK2 levels in this figure corresponds to the ratio of CDK2 band over the actin control in the blot. Combination of Taxol and CRIF1 inhibitor (3rd column) is the same case that the quantification level corresponds to the CDK2 blot over the corresponding Actin blot.

  • Please plot Fig. S2 the same way as Fig. 1

Now we have modified the plot of Fig.S2, with Figure name after the figure.